# Learning to Play With Intrinsically-Motivated, Self-Aware Agents

**Nick Haber**[1,2,3,*], **Damian Mrowca**[4,*], **Stephanie Wang**[4] , **Li Fei-Fei**[4] , and
**Daniel L. K. Yamins**[1,4,5]

Departments of Psychology[1], Pediatrics[2], Biomedical Data Science[3], Computer Science[4], and Wu
Tsai Neurosciences Institute[5], Stanford, CA 94305

{nhaber, mrowca}@stanford.edu

## Abstract

Infants are experts at playing, with an amazing ability to generate novel structured
behaviors in unstructured environments that lack clear extrinsic reward signals.
We seek to mathematically formalize these abilities using a neural network that
implements curiosity-driven intrinsic motivation. Using a simple but ecologically
naturalistic simulated environment in which an agent can move and interact with
objects it sees, we propose a "world-model" network that learns to predict the
dynamic consequences of the agent's actions. Simultaneously, we train a separate
explicit "self-model" that allows the agent to track the error map of its world-
model. It then uses the self-model to adversarially challenge the developing
world-model. We demonstrate that this policy causes the agent to explore novel
and informative interactions with its environment, leading to the generation of a
spectrum of complex behaviors, including ego-motion prediction, object attention,
and object gathering. Moreover, the world-model that the agent learns supports
improved performance on object dynamics prediction, detection, localization and
recognition tasks. Taken together, our results are initial steps toward creating
flexible autonomous agents that self-supervise in realistic physical environments.

## 1  Introduction

Truly autonomous artificial agents must be able to discover useful behaviors in complex environments
without having humans present to constantly pre-specify tasks and rewards. This ability is beyond
that of today's most advanced autonomous robots. In contrast, human infants exhibit a wide range of
interesting, apparently spontaneous, visuo-motor behaviors — including navigating their environment,
seeking out and attending to novel objects, and engaging physically with these objects in novel and
surprising ways [4, 9, 13, 15, 20, 21, 44]. In short, young children are excellent at playing —
"scientists in the crib" [13] who create, intentionally, events that are new, informative, and exciting to
them [9, 42]. Aside from being fun, play behaviors are an active learning process [40], driving self-
supervised learning of representations underlying sensory judgments and motor planning [4, 15, 24].

But how can we use these observations on infant play to improve artificial intelligence? AI theorists
have long realized that playful behavior in the absence of rewards can be mathematically formalized
via loss functions encoding intrinsic reward signals, in which an agent chooses actions that result
in novel but predictable states that maximize its learning [38]. These ideas rely on a virtuous cycle
in which the agent actively self-curricularizes as it pushes the boundaries of what its world-model-
prediction systems can achieve. As world-modeling capacity improves, what used to be novel
becomes old hat, and the cycle starts again.

---

[*]Equal contribution

Here, we build on these ideas using the tools of deep reinforcement learning to create an artificial agent that learns to play. We construct a simulated physical environment inspired by infant play rooms, in which an agent can swivel its head, move around, and physically act on nearby visible objects (Fig. 1). Akin to challenging video game tasks [26], informative interactions in this environment are possible, but sparse unless actively sought by the agent. However, unlike most video game or constrained robotics environments, there is no extrinsic goal to constrain the agent's action policy. The agent has to learn about its world, and what is interesting in it, for itself.

In this environment, we describe a neural network architecture with two interacting components, a *world-model* and a *self-model*, which are learned simultaneously. The world-model seeks to predict the consequences of agent's actions, either through forward or inverse dynamics estimation. The self-model learns explicitly to predict the errors of the world-model. The agent then uses the self-model to choose actions that it believes will adversarially challenge the current state of its world-model.

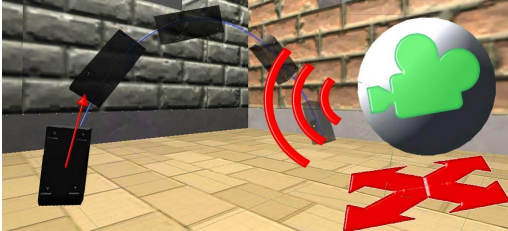

Figure 1: *3D Physical Environment.* The agent can move around, apply forces to visible objects in close proximity, and receive visual input.

Our core result is the demonstration that this intrinsically-motived self-aware architecture stably engages in a virtuous reinforcement learning cycle, spontaneously discovering highly nontrivial cognitive behaviors — first understanding and controlling self-generated motion of the agent ("ego-motion"), and then selectively paying attention to, and eventually organizing, objects. This learning occurs through an emergent active self-supervised process in which new capacities arise at distinct "developmental milestones" like those in human infants. Crucially, it also learns visual encodings with substantially improved transfer to key visual scene understanding tasks such as object detection, localization, and recognition and learns to predict physical dynamics better than a number of strong baselines. This is to our knowledge the first demonstration of the efficacy of active learning of a deep visual encoding for a complex three-dimensional environment in a purely self-supervised setting. Our results are steps toward mathematically well-motivated, flexible autonomous agents that use intrinsic motivation to learn about and spontaneously generate useful behaviors for real-world physical environments.

**Related Work** Our work connects to a variety of existing ideas in self-supervision, active learning, and deep reinforcement learning. Visual learning can be achieved through self-supervised auxiliary tasks including semantic segmentation [18], pose estimation [29], solving jigsaw puzzles [32], colorization [46], and rotation [43]. Self-supervision on videos frame prediction [23] is also promising, but faces the challenge that most sequences in recorded videos are "boring", with little interesting dynamics occurring from one frame to the next.

In order to encourage interesting events to happen, it is useful for an agent to have the capacity to select the data that it sees in training. In active learning, an agent seeks to learn a supervised task using minimal labeled data [12, 40]. Recent methods obtain diversified sets of hard examples [8, 39], or use confidence-based heuristics to determine when to query for more labels [45]. Beyond selection of examples from a pre-determined set, recent work in robotics [2, 7, 10, 36] study learning tasks with interactive visuo-motor setups such as robotic arms. The results are promising, but largely use random policies to generate training data without biasing the robot to explore in a structured way.

Intrinsic and extrinsic reward structures have been used to learn generic "skills" for a variety of tasks [6, 28, 41]. Houthooft et al. [19] demonstrated that reasonable exploration-exploitation trade-offs can be achieved by intrinsic reward terms formulated as information gain. Frank et al. [11] use information gain maximization to implement artificial curiosity on a humanoid robot. Kulkarni et al. [26] combine intrinsic motivation with hierarchical action-value functions operating at different temporal scales, for goal-driven deep reinforcement learning. Achiam and Sastry [1] formulate surprise for intrinsic motivation as the KL-divergence of the true transition probabilities from learned model probabilities. Held et al. [16] use a generator network, which is optimized using adversarial training to produce tasks that are always at the appropriate level of difficulty for an agent, to automatically produce a curriculum of navigation tasks to learn. Jaderberg et al. [22] show that target tasks can be improved by using auxiliary intrinsic rewards.

Oudeyer and colleagues [14, 33, 34] have explored formalizations of curiosity as maximizing prediction-ability change, showing the emergence of interesting realistic cognitive behaviors from simple intrinsic motivations. Unlike this work, we use deep neural networks to learn the world-model and generate action choices, and co-train the world-model and self-model, rather than pre-training the world-model on a separate prediction task and then freezing it before instituting the curious exploration policy. Pathak et al. [35] uses curiosity to antagonize a future prediction signal in the latent space of a inverse dynamics prediction task to improve learning in video games, showing that intrinsic motivation leads to faster floor-plan exploration in a 2D game environment. Our work differs in using a physically realistic three-dimensional environment and shows how intrinsic motivation can lead to substantially more sophisticated agent-object behavior generation (the "playing"). Underlying the difference between our technical approach is our introduction of a self-model network, representing the agent's awareness of its own internal state. This difference can be viewed in RL terms as the use of a more explicit model-based architecture in place of a model-free setup.

Unlike previous work, we show the learned representation transfers to improved performance on analogs of real-world visual tasks, such as object localization and recognition. To our knowledge, a self-supervised setup in which an explicitly self-modeling agent uses intrinsic motivation to learn about and restructure its environment has not been explored prior to this work.

## 2   Environment and Architecture

**Interactive Physical Environment.** Our agent is situated in a physically realistic simulated *environment* (black in Fig. 2) built in Unity 3D (Fig. 1). Objects in the environment interact according to Newtonian physics as simulated by the PhysX engine [5]. The agent's avatar is a sphere that swivels in place, moves around, and receives RGB images from a forward-facing camera (as in Fig. 1). The agent can apply forces and torques in all three dimensions to any objects that are both in view and within a fixed *maximum interaction distance* $\delta$ of the agent's position. We say that such an object is in a *play state*, and that a state with such an object is a *play state*. Although the floor and walls of the environment are static, the agent and objects can collide with them. The agent's action space is a subset of $\mathbb{R}^{2+6N}$. The first 2 dimensions specify ego-motion, restricting agent movement to forward/backward motion $v_{fwd}$ and horizontal planar rotation $v_\theta$, while the remaining $6N$ dimensions specify the forces $f_x, f_y, f_z$ and torques $\tau_x, \tau_y, \tau_z$ applied to $N$ objects sorted from the lower-leftmost to the upper-rightmost object relative to the agent's field of view. All coordinates are bounded by constants and normalized to be within $[-1, 1]$ for input into models and losses. In this setup, both the observation space (images from the 3d rendering) and action space (ego-motion and object force application) are continuous and high-dimensional, complicating the challenges of learning the visual encoding and action policy.

**Agent Architecture.** Our agent consists of two simultaneously-learned components: a *world-model* and a *self-model* (Fig. 2). The world-model seeks to solve one or more *dynamics prediction problems* based on inputs from the environment. The self-model seeks to estimate the world-model's losses for several timesteps into the future, as a function both of visual input and of potential agent actions. An action choice policy based on the self-model's output chooses actions that are "interesting" to the world-model. In this work, we choose perhaps the simplest such motivational mechanism, using policies that try to maximize the world-model's loss. In part as a review of the key issues of prediction error-based curiosity [33–35, 38], we now formalize these ideas mathematically.

*World-Model:* At the core of our architecture is the world-model — e.g. the neural network that attempts to learn how dynamics in the agent's environment work, especially in reaction to the agent's own actions. Finding the right dynamics prediction problem(s) to set as the agent's world-modeling goal is a nontrivial challenge.

Consider a partially observable Markov Decision Process (POMDP) with state $s_t$, observation $o_t$, and action $a_t$. In our agent's situation, $s_t$ is the complete information of object positions, extents, and velocities at time $t$; $o_t$ is the images rendered by the agent-mounted camera; and $a_t$ is the agent's applied ego-motion, forces and torques vector. The rules of physics are the dynamics which generate $s_{t+1}$ from $s_t$ and $a_t$. Agents make decisions about what action to take at each time, accumulating histories of observations and actions. Informally, a *dynamics prediction problem* is a pairing of complementary subsets of data — "inputs" and "outputs" — generated from the history. The goal of the agent is to learn a map from inputs to outputs. More precisely, adopting the notation $o_{t_1:t_2} = (o_{t_1}, o_{t_1+1}, \ldots o_{t_2})$ and similarly for actions and states, we let D (the "data") be fixed-time-length segments of history $\{d_t = (o_{t-b:t+f}, a_{t-b:t+f}) \mid t = 1, 2, 3 \ldots\}$. A dynamics prediction

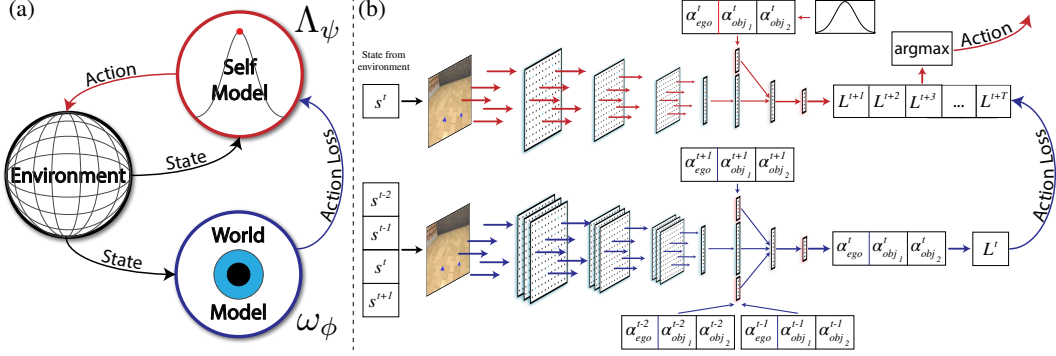

Figure 2: **Intrinsically-motivated self-aware agent architecture.** The world-model (blue) solves a dynamics prediction problem. Simultaneously a self-model (red) seeks to predict the world-model's loss. Actions are chosen to antagonize the world-model, leading to novel and surprising events in the environment (black). (a) Environment-agent loop. (b) Agent information flow.

problem (Figure 3) is then defined by specifying (possibly time-varying) maps $\iota_t : D \to$ In and $\tau_t : D \to$ Out for some specified input and output spaces In and Out, forming a triangular diagram.

Also given as part of the dynamics prediction problem is a loss $L$ for comparing ground-truth versus estimated outputs. The agent's *world-model* at time $t$ is a map $\omega_{\theta_t} : $ In $\to$ Out whose parameters are updated by stochastic gradient descent in order to lower $L$. In words, the agent's world-model (blue in Fig. 2) tries to learn to reconstruct the true-value from the input datum. Note that

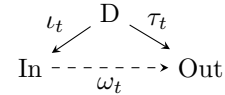

Figure 3: Diagramming dynamics prediction problems.

batches of data on which this update occurs are not drawn from any fixed distribution since they come from the history of an agent as it executes its policy, and hence this learning process does not correspond to a traditional statistical learning optimization.

Since we are focused on agents learning from an environment without external input, the maps $\iota$ and $\tau$ should in general be easy for the agent to estimate at low cost from its "sense data" — what is sometimes called self-supervision [18, 23, 29, 32, 43, 46]. For example, perhaps the most natural dynamics problem to assign to the agent as the goal of its world-model is *forward dynamics prediction*, with input $(o_{t-b:t}, a_{t-b:t+f-1})$ and true-value $o_{t+1:t+f}$. In words, the agent is trying to predict the next (several) observation(s) given past observations and a sequence (past and present) of actions. In 3-D physical domains such as ours, the outputs correspond to $f$ bitmap image arrays of future frames, and the loss function $L_F$ may be $\ell_2$ loss on pixels or some discretization thereof. Despite recent progress on the frame prediction problem [10, 23], it remains quite challenging, in part because the dimensionality of the true-value space is so large.

In practice, it can be substantially easier to solve *inverse dynamics prediction*, with input $(o_{t-b:t+f}, a_{t-b:t-1}, a_{t+1:t+f-1})$ and true-value $a_t$. In other words, the agent is trying to "post-dict" the action needed to have generated the observed sequence of observations, given knowledge of its past and future actions. Here, the loss function $L_{ID}$ is computed on (what is generally the comparitively low-dimensional) action space, a problem that has proven tractable [2, 3].

One major concern in intrinsic motivation, in particular when the agent's policy attempts to maximize the world-model's loss, is when the dynamics prediction problem is inherently unpredictable. This is sometimes referred to (perhaps in less generality than in what we proceed to define) as the *white-noise problem* Pathak et al. [35], Schmidhuber [38]. In cases where the agent's policy attempts to maximize the world-model's loss, the agent is motivated to fixate on the unlearnable. Within the above framework, this problem manifests in that there is no requirement that $\iota_t$ and $\tau_t$ actually induce a well-defined mapping In $\to$ Out that makes the diagram above commute. We refer to the existence of policies for which there are obstructions to such a commuting diagram, with nonzero probability, as *degeneracy* in the dynamics prediction problem. In fact, the inverse dynamics problem can suffer from substantial degeneracy. Consider the case of an agent pressing an object straight into the ground: no matter what the downward force is, the object does not move, so the vision and action input information is insufficient to determine the true-value.

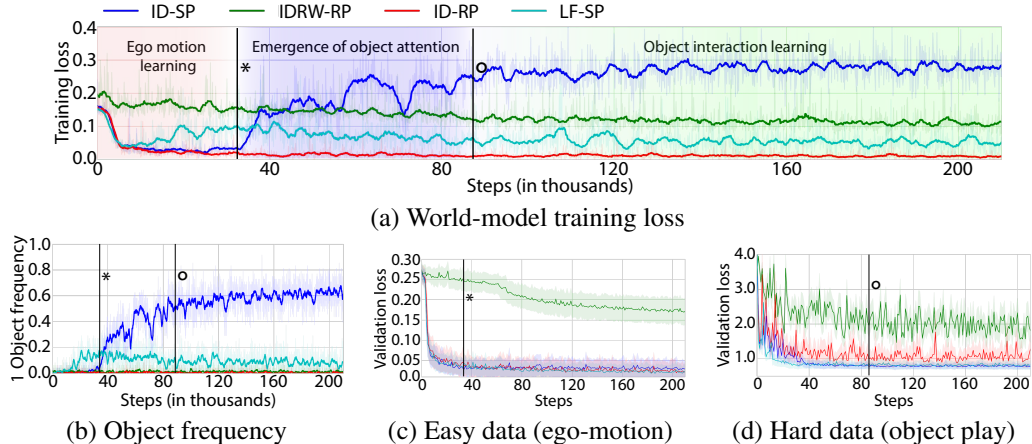

Figure 4: **Single-object experiments.** (a) World-model training loss. (b) Percentage of frames in which an object is present. (c) World-model test-set loss on "easy" ego-motion-only data, with no objects present. (d) World-model test-set loss on "hard" validation data, with object present, where agent must solve object physics prediction. Validation datasets are evaluated every 1000 batch update steps. Lighter curves represent unsmoothed batch-mean values — min and max for validation.

To avoid both of these pixel space and degeneracy difficulties, one can instead try forward dynamics prediction, but in a latent space — for example, the latent space determined by an encoder for inverse dynamics problem [35]. In this case, we begin with a system solving the inverse dynamics prediction problem and assume that its parametrization of world-model factors into a composition $\omega_{\theta_t}^{ID} = d_{\beta_t}^{ID} \circ e_{\alpha_t}^{ID}$ where $\alpha_t$ and $\beta_t$ are non-overlapping sets of parameters. We call $e_t^{ID} = e_{\alpha_t}^{ID}$ the *encoding* and the range of $e_t^{ID}$ the *latent space* $\mathcal{L}$ of the ID problem. On top, we define (time-varying) $\iota_t^{LF}, \tau_t^{LF}$ as the 1-time-step future prediction problem on trajectories in $\mathcal{L}$ given by the time-varying encoding, i.e. by $\iota_t^{LF}(d_t) = (e_t^{ID}(o_{t-b:t}), a_{t-b:0})$ and $\tau_t^{LF}(d_t) = e_t^{ID}(o_{t+1})$. The problem is then supervised by $\ell_2$ loss. The inverse-prediction world-model $\omega^{ID}$ and latent-space world-model $\omega^{LF}$ evolve simultaneously. If $\mathcal{L}$ is sufficiently low dimensional, this may be a good compromise task that represents only "essential" features for prediction.

In this work, we explore both inverse dynamics and latent space future prediction tasks.

*Explicit Self-Model:* In the strategy outlined above, the agent's action policy goal is to antagonize its world-model. If the agent explicitly predicts its own world-model loss $L_{\omega_t}$ incurred at future timesteps as a function of visual input and current action, a simple antagonistic policy could simply seek to maximize $L_{\omega_t}$ over some number of future timesteps. Embodying this idea, the self-model $\Lambda$ (red in Fig. 2) is given $o_{t-1:t}$ and a proposed next action $a$ and predicts $\Lambda_{o_{t-1:t}}(a) = (p_1(c \mid o_{t-1:t}, a) \ldots p_T(c \mid o_{t-1:t}, a))$, where $p_i(c \mid o_{t-1:t}, a)$ is the probability that the loss incurred by the world-model at time $t + i$ will equal $c$. For convenience of optimization, we discretize the losses into loss bins $C$, so that each $p_i \in \mathcal{P}(C)$ is a probability distribution over discrete classes $c \in C$. $\Lambda_{o_{t-1:t}}(a)$ is penalized with a softmax cross-entropy loss for each $i$ and averaged over $i \in 1, \ldots, T$. All future losses aside from the first one depend on future actions taken, and the self-model hence needs to predict in expectation over policy. Each $p_i(c \mid o_{t-1:t}, a)$ can be interpreted as a *map* over action space which turns out to be useful for intuitively visualizing what strategy the agent is taking in any given situation (see Fig 5).

*Adversarial Action Choice Policy:* The self-model provides, given $o_{t-1:t}$ and a proposed next action $a$, $T$ probability distributions $p_i$. The agent uses a simple mechanism to convert this data to an action choice. To summarize loss map predictions over times $t \in \{1, \ldots T\}$, we add expectation values:

$$\sigma(a)[o_{t-1:t}] = \sum_i \sum_{c \in C} c \cdot p_i(c).$$

The agent's action policy is then given by sampling with respect to a Boltzmann distribution $\pi(a \mid o_{t-1:t}) \sim \exp(\beta \sigma(\Lambda_{o_{t-1:t}}(a)))$ with fixed hyperparameter $\beta$.

*Architectures and Losses:* We use convolutional neural networks to learn both world-models $\omega_\theta$ and self-models $\Lambda_\psi$. In the experiments described below, these have an encoding structure with a common architecture involving twelve convolutional layers, two-stride max pools every other layer,

and one fully-connected layer, to encode observations into a lower-dimensional latent space, with shared weights across time. For the inverse dynamics task, the top encoding layer of the network is combined with actions $\{a_{t'} \mid t' \neq t\}$, fed into a two-layer fully-connected network, on top of which a softmax classifier is used to predict action $a_t$. For the latent space future prediction task, the top convolutional layer of $\omega_{\theta_{ID}}^{ID}$ is used as the latent space $\mathcal{L}$, and the latent model $\omega_{\theta_{LF}}^{LF}$ is parametrized by a fully-connected network that receives, in addition to past encoded images, past actions. In the ID-only case (ID-SP), we optimize $\min_{\theta_{ID}} L_{ID} + \min_{\psi} L_{\Lambda,ID}$. In the LF case (LF-SP), we optimize $\min_{\theta_{ID}} L_{ID} + \min_{\theta_{LF}} L_{LF} + \min_{\psi} L_{\Lambda,LF}$. See supplementary for details.

## 3 Experiments

We randomly place the agent in a square 10 by 10 meter room, together with up to two other objects with which the agent can interact, setting the maximum interaction distance $\delta$ to 2 meters. The objects are drawn from a set of 16 distinct geometric shapes, e.g. cones, cylinders, cuboids, pyramids, and spheroids of various aspect ratios. Training is performed using 16 asynchronous simulation instances [30], with different seeds and objects. The scene is reinitialized periodically, with time of reset randomly chosen between $2^{13}$ to $2^{15}$ steps. Each simulation maintains a data buffer of 250 timesteps to ensure stable training [27]. For model updates two examples are randomly sampled from each of the 16 simulation buffers to form a batch of size 32. Gradient updates are performed using the Adam algorithm [25], with an initial learning rate of 0.0001. See the supplement for tests of the stability of all results to variations in interaction radius $\delta$, room size, and agent speed, as well as per-object-type behavioral breakdowns.

For each experiment, we evaluate the agents' abilities with three types of metrics. We first measure the *(i) spontaneous emergence of novel behaviors*, involving the appearance of highly structured but non-preprogrammed events such as the agent attending to and acting upon objects (rather than just performing mere self-motion), engaging in directed navigation trajectories, or causing interactions between multiple objects. Finding such emergent behaviors indicates that the curiosity-driven policies generate qualitatively novel scenarios in which the agent can push the boundaries of its world-model. For each agent type, we also evaluate *(ii) improvements in dynamic task prediction* in the agents' world-models, on challenging held-out validation data constructed to test learning about both ego-motion dynamics and object physical interactions. Finding such improvements indicates that the data gleaned from the novel scenarios uncovered by intrinsic motivation actually does improve the agents' world-modeling capacities. Finally, we also evaluate *(iii) task transfer*, the ability of the visual encoding features learned by the curious agents to serve as a general basis for other useful visual tasks, such as object recognition and detection.

*Control models:* In addition to the two curious agents, we study several ablated models as controls. *ID-RP* is an ablation of ID-SP in which the world-model trains but the agent executes a random policy, used to demonstrate the difference an active policy makes in world-model performance and encoding. *IDRW-SP* is an ablation of ID-SP in which the policy is executed as above but with the encoding portion of the world-model frozen with random weights. This control measures the importance of having the action policy inform the deep internal layers of the world model network. *IDRW-RP* combines both ablations.

### 3.1 Emergent behaviors

Using metrics inspired by the developmental psychology literature, we quantify the appearance of novel structured behaviors, including attention to and acting on objects, navigation and planning, and ability to interact with multiple objects. In addition to sharp stage-like transitions in world-model loss and self-model evaluations, to quantify these behaviors we measure play state frequency and (in the case of multiple objects) the average distance between the agent and objects. We compute these quantities by averaging play state count and distance between objects, respectively, over the three simulation steps per batch update. Quantities presented below are aggregates over all 16 simulation instances unless otherwise specified.

**Object attention**. Fig. 4a shows the total training loss curves of the ID-SP, LF-SP models and baselines. Upon initialization, all agents start with behaviors indistinguishable from the random policy, choosing largely self-motion actions and rarely interacting with objects. For learned-weight agents, an initial loss decrease occurs due to learning of ego-motion, as seen in Fig. 4a. For the curious agents, this initial phase is robustly succeeded by a second phase in which loss increases. As shown in Fig. 4b, this loss increase corresponds to the emergence of object attention, in which

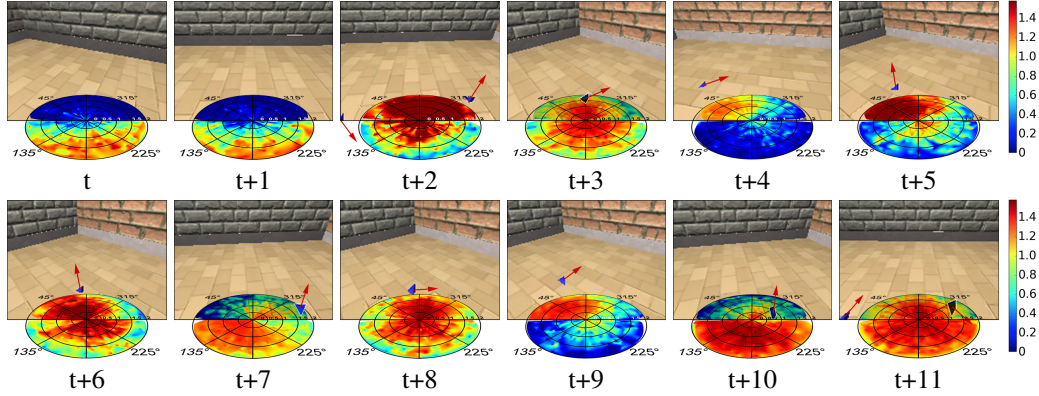

Figure 5: **Navigation and planning behavior.** Example model roll-out for 12 consecutive timesteps. Red force vectors on the objects depict the actions predicted to maximize world-model loss. Ego-motion self-prediction maps are drawn at the center of the agents position. Red colors correspond to high and blue colors to low loss predictions. The agent starts without seeing an object and predicts higher loss if it turns around to explore for an object. The self-model predicts higher loss if the agent approaches a faraway object or turns towards a close object to keep it in view.

the agent dramatically increases the play state frequency. As seen by comparing Fig. 4c-d, object interactions are much harder to predict than simple ego-motions, and thus are enriched by the curious policy: for the ID-SP agent, object interactions increase to about 60 % of all frames. In comparison, frequency of object interaction increases much less or not at all for control policy agents.

**Navigation and planning**. The curiousity-driven agents also exhibit emergent navigation and planning abilities. In Fig. 5 we visualize ID-SP self-prediction maps projected onto the agent's position for the one-object setup. The maps are generated by uniformly sampling 1000 actions $a$, evaluating $\Lambda_{o_{t-1:t}}(a)$ and applying a post-processing smoothing algorithm. We show an example sequence of 12 timesteps. The self-prediction maps show the agent predicting a higher loss (red) for actions moving it towards the object to reach a play state. As a result, the intrinsically-motivated agents learn to take actions to navigate closer to the object.

**Multi-object interactions**. In experiments with multiple objects present, initial learning stages mirror those for the one object experiment (Fig. 6a) for both ID-SP and LF-SP. The loss temporarily decreases as the agent learns to predict its ego-motion and rises when its attention shifts towards objects, which it then interacts with. However, for ID-SP agents with sufficiently long time horizon (e.g. $T = 40$), we robustly observe the emergence of an additional stage in which the loss increases further. This stage corresponds to the agent gathering and "playing" with two objects simultaneously, reflected in a sharp increase in two-object play state frequency (Fig. 6c), and a decrease in the average distance between the agent and the both objects (Fig. 6d). We do not observe this additional stage either for ID-SP of shorter time horizon (e.g. $T = 5$) or for the LF-SP model even with longer horizons. The ID-SP and LF-SP agents both experience two object play slightly more often than the ID-RP baseline, having achieved substantial one object play time. However, only the ID-SP agent has discovered how to take advantage of the increased difficulty and therefore "interestingness" of two object configurations (compare blue with green horizontal line in Fig. 6a).

## 3.2 Dynamics prediction tasks

We measure the inverse dynamics prediction performance on two held-out validation subsets of data generated from the uncontrolled background distribution of events: (i) an *easy* dataset consisting solely of ego-motion with no play states, and (ii) a *hard* dataset heavily enriched for play states, each with 4800 examples. These data are collected by executing a random policy in sixteen simulation instances, each containing one object, one for each object type. The hard dataset is the set of examples for which the object is in a play state immediately before the action to be predicted, and the easy dataset is the complement of this. This measures active learning gains, assessing to what extent the agent self-constructs training data for the hard subset while retaining performance on the easy dataset.

**Ego-motion learning**. All aside from random-encoding agents IDRW-RP and IDRW-SP learn ego-motion prediction effectively. The ID-RP model quickly converges to a low loss value, where it remains from then on, having effectively learned ego-motion prediction without an antagonistic

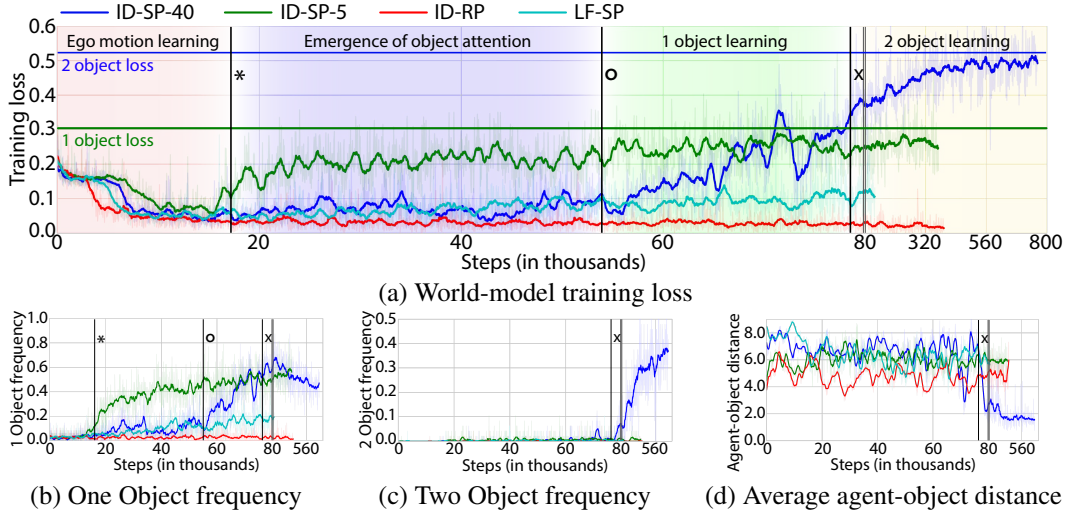

(a) World-model training loss

(b) One Object frequency     (c) Two Object frequency     (d) Average agent-object distance

Figure 6: *Two-object experiments.* (a) World-model training loss. (b) Percentage of frames in which one object is present. (c) Percentage of frames in which two objects are present. (d) Average distance between agent and objects in Unity units. For this average to be low ($\sim 2$.) both objects must be close to the agent simultaneously. Lighter curves represent unsmoothed batch-mean values.

policy since ego-motion interactions are common in the background random data distribution. The ID-SP and LF-SP models also learn ego-motion effectively, as seen in the initial decrease of their training losses (Fig. 4a) and low loss on the easy ego-motion validation dataset (Fig. 4c).

**Object dynamics prediction**. Object attention and navigation lead SP agents to substantially different data distributions than baselines. We evaluate the inverse dynamics prediction performance on the held-out hard object interaction validation set. Here, the ID-SP and LF-SP agents outperform the baselines on predicting the harder object interaction subset by a significant margin, showing that increased object attention translates to improved inverse dynamics prediction (see Fig. 4d and Table 1). Crucially, even though ID-SP and LF-SP have substantially decreased the fraction of time spent on ego-motion interactions (Fig. 4c), they still retain high performance on that easier sub-task.

### 3.3 Task transfers

We measure the agents' abilities to solve visual tasks for which they were not directly trained, including (i) object presence, (ii) localization, as measured by pixel-wise 2D centroid position, and (iii) 16-way object category recognition. We collect data with a random policy from sixteen simulation instances (each with one object, one for each object). For object presence, we subselect examples so as to have an equal number with and without an object in view. For localization and category identity (discerning which of the sixteen objects is in view), we take only frames with the object in a play state. These data are split into train (16000 examples), validation (8000 examples), and test (8000 examples) sets. On train sets, we fit elastic net regression/classification for each layer of both world- and self-model encodings, and we use validation sets to select the best-performing model per agent type. These best models are then evaluated on the test sets. Note that the test sets contain substantial variation in position, pose and size, rendering these tasks nontrivial. Self-model driven agents substantially outperform alternatives on all three transfer tasks, As shown in Table 1, the SP ($T = 5$) agents outperform baselines on inverse dynamics and object presence metrics, while ID-SP outperforms LF-SP on localization and recognition. Crucially, the ID-RP ablation comparison shows that *without an active learning policy, the encoding learned performs comparitively poorly on transfer tasks.* Interestingly, we find that training with two objects present improves recognition transfer performance as compared to one object scenarios, potentially due to the greater complexity of two-object configurations (Table 1). This is especially notable for the ID-SP ($T = 40$) agent that constructs a substantially increased percentage of two-object events.

## 4 Discussion

We have constructed a simple self-supervised mechanism that spontaneously generates a spectrum of emergent naturalistic behaviors via an active learning process, experiencing "developmental

Table 1: **Performance comparison.** Ego-motion ($v_{fwd}$, $v_\theta$) and interaction ($f$, $\tau$) accuracy in % is compared for play and non-play states. Object frequency, presence and recognition are measured in % and localization in mean pixel error. Models are trained with one object per room unless stated.

| TASK | IDRW-RP | IDRW-SP | ID-RP | ID-SP | LF-SP |
|---|---|---|---|---|---|
| $v_{fwd}$ ACCURACY — EASY | 65.9 | 56.0 | **96.0** | 95.3 | 95.3 |
| $v_\theta$ ACCURACY — EASY | 82.9 | 75.2 | **98.7** | 98.4 | 98.5 |
| $v_{fwd}$ ACCURACY — HARD | 62.4 | 69.2 | 90.4 | **95.9** | 95.4 |
| $v_\theta$ ACCURACY — HARD | 79.0 | 80.0 | 95.5 | **98.2** | 98.1 |
| $f$ ACCURACY — HARD | 20.8 | 33.1 | 42.1 | **51.1** | 45.1 |
| $\tau$ ACCURACY — HARD | 20.9 | 32.1 | 41.3 | **43.2** | 43.2 |
| OBJECT FREQUENCY | 0.50 | 47.9 | 0.40 | **61.1** | 12.8 |
| OBJECT PRESENCE error | 4.0 | 3.0 | 0.92 | 0.92 | **0.60** |
| LOCALIZATION error [PX] | 15.04 | 10.14 | 5.94 | **4.36** | 5.94 |
| RECOGNITION accuracy | 13.0 | 21.99 | 12.3 | **28.5** | 18.7 |
| RECOGNITION ACC. – 2 OBJECT TRAINING | 12.0 | - | 16.1 | **39.7** | 21.1 |

milestones" of increasing complexity as the agent learns to "play". The agent first learns the dynamics of its own motion, gets "bored", then shifts its attention to locating, moving toward, and interacting with single objects (∗ in Fig. 4 and Fig. 6). Once these are better understood (∘ in Fig. 4 and Fig. 6), the agent transitions to gathering multiple objects and learning from their interactions (× in Fig. 6). This increasingly challenging self-generated curriculum leads to performance gains in the agent's world-model and improved transfer to other useful visual tasks on which the system never received any explicit training signal. Our ablation studies show that without this active learning policy, world-model accuracy remains poor and visual encodings transfer much less well. These results constitute a proof-of-concept that both complex behaviors and useful visual features can arise from simple intrinsic motivations in a three-dimensional physical environment with realistically large and continuous state and action spaces.

In future work, we seek to generate much more sophisticated behaviors than those seen here, including the creation of complex planned trajectories and the building of useful environmental structures. Beyond the objective of building more robustly learning

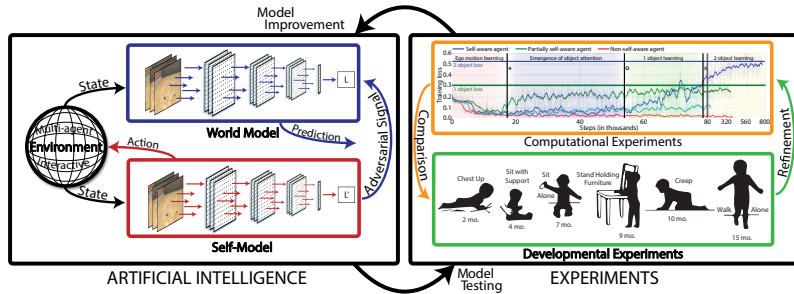

Figure 7: Computational model and human development comparison.

AI, we seek to build computational models that admit precise quantitiative comparisons to the developmental trajectories observed in human children (Figure 7). To this end, our environment will need better graphics and physics, more varied visual objects, and more realistically embodied robotic agent avatars with articulated actuators and haptic sensors. From a core algorithms approach, we will need to improve our approach to handling the inherent degeneracy ("white-noise problem") of our dynamics prediction problems. The LF-SP agent employs, as discussed in Section 2, the technique in [35] aimed at this. It was, however, unclear whether this method fully resolved the issue. It will likely be necessary to improve both the formulation of the world-model dynamics prediction tasks our agent solves as well as the antagonistic action policies of the agent's self-model. One approach may be improving our formulation of curiosity from the simple adversarial concept to include additional notions of intrinsic motivation such as learning progress [33, 34, 38]. More refined future prediction models (e.g. [31]) may also ameliorate degeneracy and lead to more sophisticated behavior. Finally, including other animate agents in the environment will not only lead to more complex interactions, but potentially also better learning through imitation [17]. In this scenario, the self-model component of our architecture will need to be not only aware of the agent itself, but also make predictions about the actions of other agents — perhaps providing a connection to the cognitive science of *theory of mind* [37].

## Acknowledgments

This work was supported by grants from the James S. McDonnell Foundation, Simons Foundation, and Sloan Foundation (DLKY), a Berry Foundation postdoctoral fellowship and Stanford Department of Biomedical Data Science NLM T-15 LM007033-35 (NH), ONR - MURI (Stanford Lead) N00014-16-1-2127 and ONR - MURI (UCLA Lead) 1015 G TA275 (LF).

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
