[Supplementary Material · supplementary.pdf]

# 5 Supplementary

In Section 5.1, we give explicit training details. In Section 5.2, we give explicit architecture details and describe the losses applied to each component. In Section 5.3, we describe and depict the objects used in the environment. In Section 5.4, we give the results of an experiment in which we halve the room's dimensions and the agents' maximum speeds, demonstrating the stability of our main results under these changes. Finally, in Section 5.5, we examine the frequencies with which the agent interacts with each distinct object.

## 5.1 Training details

Our training procedure incorporates asynchronous methods [30] and experience replay [27] with a small buffer, with data gathering threads accumulating histories of data and update threads computing gradients off of shuffled data from a buffer. We instantiate a world-model $\omega_\theta$ and self-model $\Lambda_\phi$, each with Xavier initialization. The architecture is used to collect data and world-model loss results with $N_e$ environments $\text{env}_k$ in parallel. A separate thread performs updates using data from its $N_e$ environments, syncing with the global weights, computing gradients, and updating weights with gradients.

Each data collection thread takes $\text{gather\_per\_batch}$ steps in between enqueueing a batch of size $\text{batch\_size}/N_e$. Each scene lasts a number of environment steps chosen uniformly at random within $[\text{scene\_length\_l\_bound}, \text{scene\_length\_u\_bound})$. At the beginning of each scene, objects are randomly chosen from our sixteen pairs, and the agent and object(s) are placed at positions uniformly at random in the room of size 10x10 Unity units (units defined by the Unity development platform, above referred to as "meters"). The objects are placed at random orientations just above the ground and fall at the beginning of the scene. The agent is placed upright looking in a random direction. Each maintains an history buffer $h_k$ upon which it stores observations (obtained from the environment), actions (chosen by the policy), and world-model loss (computed on data as soon as it is gathered). Policies are computed by sampling $K$ actions uniformly at random, obtaining the policy $\pi$ probabilities on each sample as described in Section 2, and sampling from this $K$-way discrete distribution. Batches are constructed from slices of data in the history buffer starting at uniformly randomly-chosen times and are placed in FIFO Queues $Q_k$.

The update thread concatenates batches dequeued from $Q_k$ for $k = 1 \ldots N_e$, and computes losses and gradients. Note that, as outlined in Section 2, different variables have different corresponding losses. For example, in the LF case, there is an auxiliary ID prediction task with variables $\theta_{ID}$ that receive gradient updates from $L_{ID}$, separately from the LF prediction task with variables $\theta_{LF}$ that receive gradient updates from $L_{LF}$. In either the ID or LF cases, the self-model has variables $\psi$ that receive gradient updates from $L_\Lambda$ which computes true-values from world-model losses stored in the data collection loop. Gradients are applied to the weights using an Adam optimizer with given $\text{learning\_rate}$.

Except where explicitly specified, we take $N_e = 16$, $\text{initial\_gather} = 250$, $\text{batch\_size} = 32$, $\text{gather\_per\_batch} = 3$ (so with 16 environments, 48 steps are taken in between each batch update), $K = 1000$, and $\text{learning\_rate} = .0001$. Despite self-model true values depending on the policy the agent chooses, we find training to be stable for small experience replay buffers of around $100 - 1000$ environment steps.

## 5.2 Model architectures and losses

We use convolutional neural networks as the base architecture to learn both world-models $\omega_\theta$ and self-models $\Lambda_\psi$. In our experiments, these networks have an encoding structure with a common architecture involving twelve convolutional layers, two-stride max pools every other layer, and one fully-connected layer, to encode all states into a lower-dimensional latent space, with shared weights across time. For the inverse dynamics task, the top encoding layer of the network is combined with actions $\{a_{t'} \mid t' \neq t\}$, fed into a two-layer fully-connected network, on top of which a softmax classifier is used to predict action $a_t$. For the latent space future prediction task, the top convolutional layer of $\omega_{\theta_{ID}}^{ID}$ is used as the latent space $\mathcal{L}$, and the latent model $\omega_{\theta_{LF}}^{LF}$ is parametrized by a fully-connected network that receives, in addition to past encoded images, past actions. See Figure 8 for a graphical representation.

**Init :**

    Dynamics prediction problem $D, \mathrm{In}, \mathrm{Out}, \iota : D \to \mathrm{In}, \tau : D \to \mathrm{Out}, L$

    World-model $\omega_\theta$

    Self-model $\Lambda_\phi$

    Environments $\mathrm{env}_k$ for $k = 1 \ldots N_e$

    batch FIFO Queues $Q_k(\text{capacity} = c)$ for $k = 1 \ldots N_e$

    History lists $h_k(\text{length} = \text{initial\_gather})$ for $k = 1 \ldots N_e$

    gather\_per\_batch, scene\_length\_l\_bound, scene\_length\_u\_bound, batch\_size

    action\_dim (8 in 1-object case, 14 in 2-object case)

    number of actions to sample $K$

    summary map $\sigma$

    learning\_rate

---

Run gather threads for each $\mathrm{env}_k$, in parallel.

**begin**

    Fill history list $h_k$ with null observations, actions, and losses.

    **while** *True* **do**

        num\_this\_batch $= 0$

        **while** num\_this\_batch $<$ gather\_per\_batch *or* total\_gathered $<$ history\_len **do**

            reset scene if needed

                **if** num\_this\_scene $\geq$ scene\_length **then**

                    observation $= \mathrm{env}_k.\text{set\_new\_scene}()$

                    delete oldest from history $h_k$

                    store observation, null action, and zero loss in $h_k$

                    num\_this\_scene $= 0$

                    scene\_length $\sim$

                      $\mathrm{Uniform}(\text{scene\_length\_l\_bound}, \text{scene\_length\_u\_bound})$

                **end**

            take an action

                **begin**

                    action\_sample $\sim \mathrm{Uniform}([-1, 1]^{K \times \text{action\_dim}})$

                    **for** $i = 0 \ldots K - 1$ **do**

                      $(p_1, p_2 \ldots p_T)[i] = \Lambda_\phi(\text{action\_sample}[i], \text{last two observations})$,

                    **end**

                    policy $\pi(i|\text{current state}) = \exp(\beta \sigma((p_1, p_2 \ldots p_T)[i]))$, normalized over $i$

                    sample i\_chosen $\sim \pi(i|\text{current state})$

                    action\_chosen $= \text{action\_sample}[i]$

                    observation $= \mathrm{env}_k.\text{step}(\text{action\_chosen})$

                **end**

            calculate $\omega_\theta$ loss

                world-model prediction $= \omega_\theta(\iota(\text{most recent history slice}))$

                world-model loss $=$

                  $L(\text{world-model prediction}, \tau(\text{most recent history slice}))$

            manage history

                delete oldest from history $h_k$

                store observation, action\_chosen, world-model loss

            num\_this\_batch $\leftarrow$ num\_this\_batch $+ 1$

            total\_gathered $\leftarrow$ total\_gathered $+ 1$

            num\_this\_scene $\leftarrow$ num\_this\_scene $+ 1$

        **end**

        store batch for update

            choose batch\_size$/N_e$ slices of $h_k$

            batch $= \iota(\text{slices}), \tau(\text{slices}), \text{world-model losses}$

            $Q.\text{enqueue}(\text{batch})$

    **end**

**end**

```
Run update thread in parallel with gather threads.
while True do
    for k = 1 ... N_e do
        | batch_k = Q_k.dequeue()
    end
    batch = concatenate(batch_k for k = 1 ... N_e)
    compute loss(es) and gradient for ω_θ (including auxiliary ID model for LF task)
    compute loss and gradient for Λ_φ using cached losses in batch
    update θ and φ with computed gradients using Adam(learning_rate)
end
```

The ID model (whether or not it is auxiliary to the LF world-model) is supervised by loss $L_{ID}$ in which we make a 3-class classification task by thresholding each dimension of the action by $-.1$ and $.1$:

$$\text{thresh}(a)_i = 1_{a_i > -.1} + 1_{a_i > .1}, i = 1 \ldots \text{action\_dim}$$

and then averaging softmax cross-entropy loss over each dimension. The LF model, if used, is supervised by $\ell_2$ loss.

The self-model is supervised by thresholding world-model losses computed in the data gathering loop (Section 5.1) by $c \in C$:

$$\text{thresh}_C(l_t) = \sum_{c \in C} 1_{l > c}$$

and averaging softmax cross-entropy loss over $T$ successive timesteps. In the 1-object setting, we took $C = \{.28\}$ for the ID-SP case and $C = \{.13\}$ for the LF-SP case, tuned for object attention (in practice, this appears to matter only in that it satisfies a constraint: below almost all 1-object play losses, for a ID-RP/LF-RP model, but above almost all ego-motion losses, between the first 10000-30000 steps). In the 2-object setting, we chose $C = \{.28, .44, .59\}$ for ID-SP and $C = \{.13, .26, .59\}$ for LF-SP.

To summarize the optimization criteria, in the ID-only case (ID-SP), two objectives are optimized:

$$\min_{\theta_{ID}} L_{ID} + \min_{\psi} L_{\Lambda, ID},$$

where $L_{ID}$ is a sum, across dimension, of softmax cross-entropy losses on 3-way discretizations of each action dimension, and $L_{\Lambda, ID}$ is a sum, across $T$ timesteps, of softmax cross-entropy losses on $C$-way discretizations of $\omega^{ID}$ loss. In the LF case (LF-SP), three objectives are optimized:

$$\min_{\theta_{ID}} L_{ID} + \min_{\theta_{LF}} L_{LF} + \min_{\psi} L_{\Lambda, LF},$$

where $L_{LF}$ is $\ell_2$-loss on the latent space and $L_{\Lambda, LF}$ is like $L_{\Lambda, ID}$ but with $\omega^{ID}$ loss replaced with $\omega^{LF}$.

### 5.3   Object details

In Figure 5.5, we depict the objects used and give a breakdown of play frequency per object. Shapes are given equal mass (1 Unity unit of mass) and are blue of the same texture. Their geometries consists of varied aspect ratios of four types of shape: sphere, cube, cone, and cylinder, with four per type.

### 5.4   Stability under varied setups.

In this section, we present results in which we vary both the environment and the agent's maximum ego-motions, demonstrating stability under this change of emergent "developmental milestones" and dynamics prediction problem performance gains under the antagonistic policy of Section 2. We make the room 5 by 5 meters (halving each dimension) and divide the agent's maximum ego-motion (forward/backward $v_{fwd}$ and planar angular $v_\theta$) by two while keeping the maximum interaction distance $\delta = 2$ fixed. The same 16 objects are placed in 16 environments, one per environment, as in the 1-object experiments of Section 3. We find (Figure 10) that the same sorts of milestones (ego-motion learning, object attention, improved object dynamics prediction) emerge, with similar comparisons to baseline, only approximately 4 times as fast. Interestingly, after some time, the

# ID

$o_{t-1:t}$, concatenated by channel

3x3 conv, 64
Relu

3x3 conv, 64

2-stride pool
Relu

3x3 conv, 64
Relu

3x3 conv, 64

2-stride pool
Relu

3x3 conv, 128
Relu

3x3 conv, 128

2-stride pool
Relu

3x3 conv, 128
Relu

3x3 conv, 128

2-stride pool
Relu

3x3 conv, 192
Relu

3x3 conv, 192

2-stride pool
Relu

3x3 conv, 192
Relu

3x3 conv, 192

2-stride pool
Relu

fc, 512
Relu

$\oplus$

$a_{t-2}$, $a_{t-1}$, $a_t$

fc, 512
Relu

fc, 3 * action_dim

$o_{t:t+1}$
shared encoding

$o_{t-2:t-1}$
$o_{t-3:t-2}$
shared encoding
shared encoding

# Self-model

$o_{t-1:t}$, concatenated by channel

3x3 conv, 64
Relu

3x3 conv, 64

2-stride pool
Relu

3x3 conv, 64
Relu

3x3 conv, 64

2-stride pool
Relu

3x3 conv, 64
Relu

3x3 conv, 64

2-stride pool
Relu

3x3 conv, 64
Relu

3x3 conv, 64

2-stride pool
Relu

3x3 conv, 64
Relu

3x3 conv, 64

2-stride pool
Relu

3x3 conv, 64
Relu

3x3 conv, 64

2-stride pool
Relu

fc, 512
Relu

action_sample

fc, 512
Relu

fc, T * |C|

# LF

$a_{t-2}$, $a_{t-1}$

$e(o_{t-2:t-1})$, $e(o_{t-1:t})$

$\oplus$

fc, 1152
Relu

fc, 1152
Relu

fc, 1152

Convolution
Max Pool
Fully-connected
Concatenate $\oplus$

Figure 8: The ID, LF, and self-model architectures.

Figure 9: The objects.

world-model loss dipped, and we hypothesize (but due to computational constraints, did not run out sufficiently long, given this 4x heuristic) that we would see this behavior in our main setup, as well.

### 5.5 Object frequency breakdown.

To measure how learned object attention depends on shape, we modify our training procedure, assigning each of our 16 environments a unique shape — each environment is assigned a single shape that it uses for each reset, so that at all points in training, each shape is in exactly one environment. The environment and agent parameters are as described in Section 5.4, differing in environment dimensions and agent speeds from our main experimental setups. We then measure object play frequency broken down by object (Figure 5.5). Note the heterogeneity — while most objects have similar play frequency graphs, others have inconsistent play frequency. This suggests that the control problem of finding an object, and keeping it in view, is not learned with equal success across objects.

Figure 10: **Single-object experiments, smaller room and speed.** (a) World-model training loss. (b) Percentage of frames in which an object is present. (c) World-model test-set loss on "easy" ego-motion-only data, with no objects present. (d) World-model test-set loss on "hard" validation data, with object present, where agent must solve object physics prediction. This experiment differs from those in the main text (compare with Figure 4) by halving both the room size and maximum ego-motion speeds while keeping the maximum interaction distance fixed.

Figure 11: Object in view frequency across all objects and for each of the tested 16 objects individually.