[Reviews · NeurIPS 2018]

Reviewer 1



This paper develops a system that explores new approaches to the problem of using active learning (or play) to learn the structure of an environment and how it can be influenced. It compares four agents (two loss functions, and three "ablated" agents to test contributions of different features, e.g., an active learning policy versus a random one), looking at loss over time and arguing that one sees state changes that are analogous to developmental changes in human learners. Despite the clarity of the prose, there was a lot in this paper that I found hard to follow, despite having looked through the supplementary information. That's due in part (but not wholly) to my limited expertise with deep learning. Perhaps most importantly, I don't know exactly how the different parts of the performance comparison (Table 1) worked, and whether the performance of the agents was actually good relative to other approaches, versus just better than the ablated alternatives. I also felt that if the target audience includes cognitive scientists and developmental psychologists, there needs to be a stronger and more explicit connection between the ideas in the introduction and the design decisions on pages 4 and 5. As for the technical details, I did not go through them in enough depth to comment one way or another. I've read the author feedback, which provided some reassurance that the baselines are "serious" and I applaud the authors' decision to clarify and expand connections between the authors' work and empirical research and theories in developmental psychology.

Reviewer 2



This is a really interesting paper that proposes algorithm for self-supervised learning. The innovation here is to import some ideas from adversarial learning and GANS to the topic of self-supervision. Specifically and authors propose and test a model where a world model of some physics domain is “challenged” by a self-model that proposes actions that are expected to maximize the loss of the world models. Both models then learn based on experience and the challenging by the self model help to guide the world model. The findings from this paper are fascinating, particularly the result that the system initially learns the dynamics of its own motion before getting “bored” and moving on to attending to objects. I think this type of result is likely to have important impact no only in computer science/AI/ML communities but also for developmental psychologists interested in how babies bootstrap their understanding of the world. If I had to make a suggestion, I would say that the main text on page 4 (describing the world model) is a little abstract. I get the authors are trying to present a very general version of the issue of learning a world model, however I found this discussion a little tangential and distracting. It seems enough to describe the specific world model approach you use in more detail rather than exploring some of the general issues here. The literature review was extensive and very well done. None the less you might find this paper interesting given some of what you were saying: https://arxiv.org/abs/1711.06351. Also for something on learning about the physical world through active exploration: https://psyarxiv.com/u9y4c . Don’t necessarily cite it just know about it! They are possibly interesting and relevant for the future directions laid out about more complex, planned trajectories and actions.

Reviewer 3



The work presents a new approach on exploration yielding towards self-supervising agents. The proposed idea relies on learning two models, a world-model and a self-model. The former predicts the world dynamics while the latter predicts errors of the world-model and is used to select actions that challenge the current world-model. This enables the agent to develop a good world model. The overall approach is interesting but as far as I understood, it delivers in the end another approach for exploration. The applied metrics are not further specified than in the paragrapg starting in line 219, so the actual evaluation is unclear to me, e.g. I do not understand how exactly the appearence of interesting behaviors is quantified. It is not clear to me how exactly the different phases are determined i.e. the onsets of 'Emergence of object attention', 'Object interaction learning' etc. It would also have been nice to see a video / simulation of the behavior over the whole period (or at least plots from important phases, similar to Figure 5). Due to the two points mentioned above (and brevity in explanation), the experiments and the respective evaluations are a bit squishy for me and i can not grasp what is actually achieved. Especially in the part of task transfer (starting in line 286) it is not clear to me what exactly was done. For example, I do not know what exactly is supposed to be transferred and also the whole setup did not appear to me to actively solve tasks but to rather explore and learn models. Thus, I also do not understand what is presented in Table 1. The accuracy is computed on what? In Figure 4, (c) and (d) have only 200 steps or is the addition (in thousands) missing? It is also not clear on how many trials every model was evaluated and thus, for (a) and (b) I'm not sure whether I am just looking at a smoothed curve plotted over the original values? For (c) and (d), it looks more definite like an error area, but it is not clear to me whether I see the variance, twice the standard deviation or whatever of how many evaluations? The same holds for Figure 6. A problem of the presented action space representation may be that it depends on the number of objects available. This may become quite hard for messy and cluttered environments. Also, the highest amount of objects evaluated was N = 2, so it would be interesting to see how well the approach scales for fuller rooms and how the behavior develops. Additionally, for me, having almost the same amount of supplementary as submitted pages is a bit too much and unfair towards other authors who are limiting themselves. Thus, an alternative option could be to publish the work as a journal article.